# Newborn Screening for Gaucher Disease: The New Jersey Experience

**DOI:** 10.3390/ijns11020034

**Published:** 2025-05-02

**Authors:** Caitlin Menello, Shaney Pressley, Madeline Steffensen, Sarah Schmidt, Helio Pedro, Reena Jethva, Karen Valdez-Gonzalez, Darius J. Adams, Punita Gupta, Lorien Tambini King, Milen Velinov, Sharon Anderson, Peyman Bizargity, Beth Pletcher, Allysa Tuite, Christina Kresge, Debra Lynn Day-Salvatore, Ryan Kuehl, Can Ficicioglu

**Affiliations:** 1Section of Biochemical Genetics, Division of Genetics, Children’s Hospital of Philadelphia, Philadelphia, PA 19104, USA; pressleyse@chop.edu (S.P.); steffensem@chop.edu (M.S.);; 2Department of Pediatrics, Perelman School of Medicine, University of Pennsylvania, Philadelphia, PA 19104, USA; 3Pediatric Genetics & Genomics, Hackensack University Medical Center at Hackensack Meridian Health, Hackensack, NJ 07601, USA; helio.pedro@hmhn.org (H.P.); karen.valdez-gonzalez@hmhn.org (K.V.-G.); 4Clinical Biochemical Genetics, Atlantic Health System, Morristown, NJ 07960, USA; darius.adams@atlantichealth.org; 5Medical Genetics, St. Joseph’s University Medical Center, Paterson, NJ 07503, USA; gupitap@sjhmc.org (P.G.); kinglorie@sjhmc.org (L.T.K.); 6Medical Genetics, Rutgers Health, Rutgers Robert Wood Johnson Medical School, New Brunswick, NJ 08901, USA; mv662@rwjms.rutgers.edu (M.V.); sharon.anderson@rutgers.edu (S.A.); 7Clinical Genetics, University Hospital-Newark at Rutgers Health, Newark, NJ 07103, USA; pb752@njms.rutgers.edu (P.B.); pletchba@njms.rutgers.edu (B.P.); allygo@njms.rutgers.edu (A.T.); bondcm@njms.rutgers.edu (C.K.); 8Department of Medical Genetics and Genomic Medicine, St. Peter’s University Hospital, New Brunswick, NJ 08901, USA; ddaysalvatore@saintpetersuh.com (D.L.D.-S.); rkuehl@saintpetersuh.com (R.K.)

**Keywords:** Gaucher disease, lysosomal storage disorder, newborn screening

## Abstract

Gaucher disease (GD) is a lysosomal storage disorder (LSD) characterized by glycosphingolipid accumulation. Age of symptomonset and disease progression varies across types of disease. Newborn screening (NBS) for Gaucher disease facilitates early identification of affected individuals and enables pre-symptomatic monitoring with the goal of starting therapies early and improving clinical outcomes. This multi-center study involved New Jersey NBS referral centers. Data regarding initial NBS results, confirmatory testing, diagnosis, and treatment were collected. For patients on therapy, monitoring biomarkers and exam findings are available as of the last clinical evaluation. Between July 2019 and December 2023, 438,515 newborns were screened, with 60 screen-positive cases. Of those positive screens, 19 cases with positive screens did not undergo confirmatory testing due to parental refusal, loss to follow-up, or death; 23 cases were false positives; 14 newborns were diagnosed with GD type I; 2 newborns were diagnosed with suspected type I GD; 2 newborns were diagnosed with GD type II; and 1 case is still pending. Three type I GD patients started enzyme replacement therapy, with the youngest starting at 28 months of age. Post-treatment data are available for these individuals. One type II case was referred to experimental gene therapy, and one was started on ERT. Our results demonstrate that NBS for GD is a valuable public health tool that can facilitate early diagnosis and intervention.

## 1. Introduction

Gaucher disease (GD) is a multi-systemic lysosomal storage disorder (LSD) caused by pathogenic biallelic variants in GBA and resulting in deficiency of β-glucosidase activity [1,2]. Consequent glycosphingolipid accumulation results in clinical manifestations of disease [3,4].

Gaucher disease is differentiated into three main forms based on the presence and rate of progression of neurologic disease [1,5,6]. Type I GD, considered the non-neuronopathic form, is characterized by visceral symptoms without primary neurologic involvement. Individuals with type I disease may present at any age with hepatosplenomegaly, periodic pain crises, bone disease, respiratory disease, cytopenia, anemia, and poor growth [6,7,8]. Type II GD typically presents in the first year of life and is characterized by rapidly progressive neurologic involvement, as well as marked organomegaly, cytopenia, and other visceral involvement [7,9]. Individuals with type III GD may present with similar symptoms as type I but do develop neurologic symptoms, including cognitive impairments, seizures, ataxia, and oculomotor abnormalities [6,10]. The global incidence of GD is believed to be between 0.45 and 25.0/100,000 live births [11]. The incidence is higher in the Ashkenazi Jewish population, with type I GD occurring in roughly 1 in 450 live births [12]. Type I GD is the more prevalent form found in affected individuals in Western countries, including the USA [7].

In the US, enzyme replacement therapy (ERT) is approved for type I and type III GD, though it does not slow neurologic progression [7,13]. Off-label use of ERT may be utilized palliatively for type II GD to address somatic involvement [14,15]. Substrate reduction therapy is approved for patients with type I GD 18 years of age and older.

Guidelines recommend initiation of ERT with development of manifestations in pediatric populations [7,16]. However, many patients with Gaucher disease experience a diagnostic odyssey. Roughly one in six affected individuals experience a delay of diagnosis ≥7 years from the first time they present to a doctor with symptoms [17]. Delays can be due to clinical heterogeneity, non-specific symptoms, and/or misdiagnosis [17,18]. Given the progressive nature of disease such delays may lead to poorer clinical outcomes.

Newborn screening (NBS) for Gaucher disease facilitates early identification of affected individuals and enables pre-symptomatic monitoring. Monitoring using non-invasive methods such as medical history, physical exams, and biomarker testing can reliably identify individuals who require therapy. Such surveillance is improved with the identification of lyso-Gb1, also known as glucosylsphingosine, as a sensitive and specific biomarker for accurate diagnosis, monitoring of glucosylceramide accumulation, and clearance with ERT [19]. A previous study demonstrated the utility of lyso-Gb1 in differentiating between types of Gaucher disease and monitoring for treatment response in pediatric populations [20]. Notably, patients with type I disease had lower elevations or even normal levels of lyso-Gb1 compared to individuals with types II and III [20]. Therefore, lyso-Gb1 may be useful in the differentiation of Gaucher disease subtypes and in monitoring treatment response.

Gaucher disease is not on the Recommended Uniform Screening Panel (RUSP) in the US. Currently, there are six states screening for GD in all infants: Illinois, Missouri, New Jersey, Tennessee, Oregon, and New Mexico [21,22,23]. Newborn screening for Gaucher disease is offered at certain hospitals in New York and Pennsylvania [24].

New Jersey implemented mandated newborn screening for Gaucher disease in July 2019. A tiered testing approach was implemented with confirmatory enzyme activity as first-tier testing and *GBA* sequencing as second-tier testing. However, second-tier testing was discontinued in 2020 due to the COVID-19 pandemic. There are nine NBS referral centers in the state of New Jersey. Here we present multi-center data from the New Jersey NBS program and discuss clinical management of affected individuals identified through newborn screening.

## 2. Materials and Methods

### 2.1. Human Subjects Research

The Institutional Review Board at the Children’s Hospital of Philadelphia determined that this study met exemption criteria per 45 CFR 102(e). A deidentified retrospective review of New Jersey’s NBS results and outcomes from participating referral centers was performed for all cases reported from 8 July 2019 through 31 December 2023. Limited clinical data, including repeat β-glucosidase activity, glucopsychosine levels, and *GBA* sequencing and deletion/duplication analysis, were reported by participating referral centers.

### 2.2. New Jersey Newborn Screening Protocol

β-glucosidase activity is measured via tandem mass spectrometry using the NeoLSD™ MSMS Kit by Revvity, Inc. [25]. The NeoLSD™ MSMS Kit is commercially available and analyzes the activity of the six lysosomal enzymes associated with Gaucher disease, Pompe disease, Fabry disease, Krabbe disease, acid sphingomyelindase deficiency, and Mucopolysaccharidosis type I. The cutoff for the assay is expressed as the percentage of the daily median. Single enzyme deficiency of β-glucosidase is considered abnormal for Gaucher disease. If multiple enzymes are deficient, the sample is considered unsatisfactory and a repeat sample is requested.

### 2.3. Abnormal Newborn Screen Follow Up

Infants with decreased GBA enzyme on dried blood spot specimens were considered “positive”, and referral to a biochemical geneticist was recommended. Families can decline further evaluation of an abnormal NBS after discussion with the primary care provider. Parental refusal must be documented by the state. Otherwise, selection of a particular referral center was at the discretion of the infant’s primary care provider.

Screen-positive infants referred to a metabolic center were promptly evaluated by a physician and nurse practitioner or other advanced practice provider. A genetic counselor was available to provide counseling at most centers. The initial evaluation involved a physical exam and detailed collection of clinical and family history. Confirmatory β-glucosidase activity, lyso-Gb1 levels, and molecular *GBA* analysis were generally recommended at initial evaluation. Due to differences in provider preference across multiple institutions, confirmatory biochemical and molecular testing were performed at various laboratories. All laboratories that performed such testing were CLIA/CAP certified. Of the nine referral centers in the state, one declined to participate.

## 3. Results

From July 2019 to December 2023, 438,515 newborns underwent screening, resulting in 60 screen-positive cases. Of those positive screens, two cases were evaluated by specialists at external (non-referral) centers. One was reported as a false positive, while the other is pending with the state. Nineteen screen-positive newborns did not undergo confirmatory testing due to parental refusal, loss to follow-up, or death. Notably, 63% of the cases lacking confirmatory testing were attributed to parental refusal. Twenty-three screen-positive newborns were determined to be false positive cases after repeat enzyme was normal and/or *GBA* sequencing was non-diagnostic.

Out of 60 screen-positive newborns, 18 were ultimately diagnosed with GD: 14 were diagnosed with type I GD, 2 were diagnosed with suspected type I GD based on low enzyme activity and compound heterozygosity of the GBA p.N409S allele with a variant of uncertain significance, and 2 were diagnosed with type II GD (Table 1). All subjects’ NBS and confirmatory enzyme levels, genotypes, lyso-Gb1 levels, and treatment statuses are presented in Table 1.

Subjects 1, 2, and 3 started ERT at 42 months, 29 months, and 28 months of age, respectively. Notably, Subject 3’s parents initially refused further workup after discussion with the primary care physician. Subject 3 returned to care at 26 months of age after an older sibling was diagnosed with type I Gaucher disease. The older sibling had presented with a severe bone crisis and femur necrosis at age 4 years. Subjects 17 and 18 started off-label use of ERT as a palliative measure at 1 month and 13 months, respectively. Treatment information for subjects 1, 2, and 3 is presented below.

### 3.1. Treatment Initiation and Response in Type I GD

#### 3.1.1. Subject 1

After the initial evaluation and diagnosis, follow-up appointments were recommended every six months. Interim history and exams from baseline to 24 months were considered normal, though physical exams were limited due to the need for telehealth visits during the COVID-19 pandemic.

The physical exam performed at 30 months was notable for mild splenomegaly. An abdominal ultrasound performed at 34 months demonstrated mild hepatomegaly with normal spleen size. Follow-up laboratory tests performed at 36 and 41 months revealed an increase in lyso-Gb1 levels up to 13 times the upper limit of normal (Table 2). An MRI of the liver, spleen, and bone marrow with elastography was performed at 40 months. The MRI demonstrated mild hepatosplenomegaly (liver volume 1.5 times normal with liver stiffness at EPI 2.65 kPa and spleen volume 6.9 times normal with increased splenic stiffness at EPI 5.43 kPa). The MRI also noted decreased T1-weighted signal intensity in the distribution of hematopoietic marrow without signs of osteonecrosis. The patient’s family denied any other clinical symptoms in the child. Initiation of ERT was recommended.

Subject 1, aged 42 months at treatment initiation, received 60 u/kg of imiglucerase every two weeks. Infusions were tolerated well with no symptoms of hypersensitivity or anti-drug antibody development. Laboratory monitoring performed at 46 months demonstrated down-trending lyso-Gb1 level to 5 times the upper limit of normal (Table 2). Repeat MRI with elastography performed at 58 months demonstrated stable hepatosplenomegaly with liver volume 1.5 times normal and normal stiffness and a spleen volume 5.6 times normal with increased stiffness at EPI 4.05 kPa. Bone marrow findings were stable compared to the prior study. Lyso-Gb1 at the last clinical evaluation, at 62 months, decreased to 3 times the upper limit of normal (Table 2). A physical exam performed at that time noted resolution of hepatosplenomegaly. Other clinical symptoms of Gaucher disease were denied.

#### 3.1.2. Subject 2

After initial evaluation and diagnosis, follow-up was recommended every six months. Exams and interim history from baseline to 16 months were considered normal, though physical exams were limited due to the need for telehealth visits during the COVID-19 pandemic. Biomarkers at 16 months of age demonstrated a 7-fold increase in lyso-Gb1 level and an initial chitotriosidase level of >70 times the upper limit of normal (Table 3). At 21 months of age, lyso-Gb1 increased to 20 times the upper limit of normal, and chitotriosidase increased to >85 times the upper limit of normal (Table 3). An abdominal ultrasound with elastography performed at that time demonstrated normal size and compliance of both the spleen and liver. However, at 25 months his weight gain stalled, though linear growth remained normal.

The follow-up evaluation at 28 months of age was remarkable for notable fatigue and easy bleeding and bruising. Laboratory tests at that time demonstrated pancytopenia; the lyso-Gb1 level was elevated at 23 times the upper limit of normal, and chitotriosidase was stably elevated at >80 times the upper limit of normal (Table 3). The physical exam was remarkable for splenomegaly. An MRI of the liver and spleen with elastography performed at 29 months revealed liver volume 1.7 times normal with increased stiffness at EPI 3.3 kPa and spleen volume 15.3 times normal with increased stiffness at EPI 6.1 kPa.

Subject 2 was 29 months old at treatment initiation, receiving 60 u/kg of imiglucerase every two weeks. Infusions were tolerated well with no symptoms of hypersensitivity or anti-drug antibody development. Laboratory evaluation at 34 months demonstrated improvement of hematologic parameters, decreasing lyso-Gb1 at 13 times the upper limit of normal, and decreasing chitotriosidase at 44 times the upper limit of normal (Table 3). An MRI of the liver, spleen, and bone marrow with elastography was performed at 41 months and revealed liver volume 1.2 times normal with normal stiffness at EPI 1.9 kPa, as well as spleen volume 4.7 times normal with increased stiffness at EPI 7.1 kPa. Bone marrow demonstrated normal marrow signal intensity without focal abnormality. The last clinical evaluation at 49 months of age demonstrated improved biomarkers with lyso-Gb1 elevated at 4 times the upper limit of normal (Table 3). The physical exam performed at that time demonstrated resolution of splenomegaly with improvement in linear growth and weight gain.

#### 3.1.3. Subject 3

After consultation with the primary care provider, Subject 3’s parents declined a referral to a specialist for confirmatory testing. Subject 3 did not pursue any specialty care until his older sibling, who was born before the implementation of newborn screening, was diagnosed with type I GD. The older sibling had presented with a severe bone crisis and femur necrosis at age 4 years. Subject 3 presented for an initial evaluation by a specialist at 26 months of age. Hepatosplenomegaly was noted during the physical exam. Confirmatory lyso-Gb1 level was increased at 14 times the upper limit of normal, and chitotriosidase was increased at 73 times the upper limit of normal (Table 4). A complete blood count (CBC) demonstrated low hemoglobin and platelet counts (Table 4). Clinical symptoms were denied. An abdominal ultrasound at 27 months revealed hepatosplenomegaly and diffuse increased hepatic parenchymal echogenicity.

Subject 3 began treatment at 28 months of age with 60 u/kg of imiglucerase every 2 weeks. Laboratory evaluation at 30 months demonstrated a decreased lyso-Gb1 level at 4 times the upper limit of normal and a decreasing chitotriosidase level at 36 times the upper limit of normal (Table 4). At his last clinical evaluation at 34 months of age, lyso-Gb1 remained stable at 4 times the upper limit of normal, and chitotriosidase decreased to 12 times the upper limit of normal (Table 4). The physical exam performed at that time was normal, with no detectable hepatosplenomegaly. Other clinical symptoms of Gaucher disease were denied.

## 4. Discussion

Newborn screening is a public health initiative that successfully identifies children with rare, treatable disorders to enable prompt access to disease-modifying therapies. Wilson and Jungner previously published criteria for adding conditions to population screening initiatives [26]. Gaucher disease appears to be a candidate for population screening based on the following criteria: (a) a specific screening test based on enzyme analysis is available, (b) confirmatory tests, including enzyme assays, specific biomarkers, and genetic tests, are accessible, (c) disease-modifying therapies exist, (d) diagnostic delays can extend to 7 years or more in some cases, (e) the disease often has an early onset, presenting in the first years of life, and (f) irreversible bone disease, which could be prevented by early treatment, is a common presentation. These factors collectively support the consideration of Gaucher disease for population screening programs.

It is well established that early diagnosis and prompt initiation of therapy improve clinical outcomes in Gaucher disease [7,13,16]. Though some cases of type I Gaucher disease may present later in life, newborn screening enables early, non-invasive clinical monitoring for signs of disease, prevents the diagnostic odyssey, and allows early treatment.

In type II Gaucher disease, although treatment cannot prevent central nervous system (CNS) involvement, newborn screening can limit the diagnostic odyssey of affected newborns and lead to early initiation of enzyme replacement therapy (ERT) to reduce visceral disease manifestations [14,15].

The incidence of Gaucher disease across all subtypes in New Jersey was approximately 1 in 24,362 live births between July 2019 and December 2023. This incidence is higher than what was previously reported in other states, such as Illinois (1 in 43,959), Missouri (1 in 43,701), and Oregon (1 in 36,695), as well as other countries, such as China (1 in 80,855) [21,22,23,27]. Among the 18 confirmed cases of Gaucher disease within this population, pre-symptomatic monitoring enabled early identification of disease manifestations for three children within the first three and a half years of life. Early signs of disease included organomegaly, impaired growth, hematologic abnormalities, and elevated biomarkers of disease, most notably, lyso-Gb1 and chitotriosidase. Treatment was tolerated well in all three individuals, with no significant signs of hypersensitivity. ERT initiation resulted in reduction of biomarkers in all three patients. Subjects 1 and 2 demonstrated resolution of organomegaly during the follow-up period. Clinical symptoms, when reported, were resolved with therapy, and all three subjects are doing clinically well with no new reported issues. This report provides evidence that newborn screening for Gaucher disease benefits the general population by enabling pre-symptomatic diagnosis and monitoring of affected children during the latent period of this disease.

The high false positive rate in New Jersey indicates the need for improved screening methods for Gaucher disease. New Jersey newborn screening for lysosomal storage disorders, including Gaucher disease, is currently performed as a single-tier test. The false positive cases were cleared after repeat enzyme was normal and/or *GBA* sequencing was non-diagnostic. Carriers of Gaucher disease can have indeterminate or low enzyme levels on leukocyte testing. False positive cases with either negative or heterozygous pathogenic variants in *GBA* were found to have variable levels of enzyme activity reported on the newborn screen. The lowest reported enzyme activity in a false positive case was 7.4% [≥12.0%], while the highest was 11.6% [≥12.0]. However, one true positive, Subject 4, demonstrated borderline enzyme activity on newborn screening at 11.6%. This suggests that adjusting the enzyme activity cutoff may result in false negative cases. Therefore, a more nuanced approach to screening is needed to improve accuracy and reduce false positives while avoiding false negatives.

Illinois’s pilot program reported that 74% of screen-positive newborns for Gaucher disease were premature [21]. While rates of false positives in screen-positive infants who were premature were not reported, this finding raises concern that gestational age may impact the ability to interpret screening results for Gaucher disease. The gestational ages and birth weights of false positives were not written on newborn screening records in New Jersey. It is possible that both resulted in false positive results in our cohort. Implementing cutoff values based on different gestational ages and birth weights may help reduce the number of false positives.

Tiered testing would be beneficial for Gaucher disease newborn screening. This approach is already utilized for newborn screening for other lysosomal storage disorders (LSDs) such as Pompe disease, Mucopolysaccharidosis type I (MPS I), and Mucopolysaccharidosis type II (MPS II) in some states. For Gaucher disease, a three-tiered approach is recommended, with enzyme activity measurement as the first tier, *GBA* sequencing as the second tier, and lyso-Gb1 as the third tier. The development of second- and third-tier tests, such as genotype and/or biomarkers, has shown effectiveness in reducing recall rates for certain LSDs, such as Pompe, MPS I, and Gaucher disease [28,29]. Utilization of a tiered approach can aid in the prompt identification of affected newborns while reducing false positives.

For twelve of the screen-positive cases, no initial confirmatory testing was performed due to parental refusal. Refusal for additional testing occurred both after an initial discussion with the pediatrician and after an initial evaluation by a specialist. The state does not require documentation of the reason for parental refusal. However, the higher-than-expected rate of parental refusal may reflect attitudes towards newborn screening for Gaucher disease. There might be parental concern about labeling their children with a rare disease such as Gaucher in certain cultures or ethnic groups. In addition, there might be a parental perception of Gaucher disease being a late-onset disorder that does not need early diagnosis and treatment.

Gaucher disease exists as a clinical spectrum. There are individuals with type I GD who remain asymptomatic throughout their lives. For individuals with a family history of type I GD or for those who are a part of ethnic groups in which type I Gaucher disease is highly prevalent, at-risk individuals may refuse confirmatory testing if they have personal experience with affected individuals who’ve not needed treatment. Knowledge of disease or carrier status may increase fears of stigma for children identified through newborn screening. A previous study revealed parental attitudes towards newborn screening for Pompe disease [30]. Further work is needed to elucidate specific reasons for parental refusal and understand attitudes regarding newborn screening for Gaucher disease in the general population.

Newborn screening for Gaucher disease has the potential to significantly improve clinical outcomes for affected children by enabling early diagnosis. Clinical monitoring using non-invasive methods can effectively identify affected individuals who need therapy. However, comprehensive guidelines for monitoring during the pre-symptomatic period are still needed. Methods to improve perinatal education regarding newborn screening for LSDs, as well as culturally competent counseling strategies, are needed to ensure families understand the implications of a positive newborn screen and can make informed decisions regarding follow-up.

To enhance the effectiveness of newborn screening for Gaucher disease, several improvements are necessary: (a) Developing more effective methods to educate expectant parents about newborn screening for Gaucher disease is crucial. This education should begin during prenatal care and continue through the immediate postnatal period. (b) Implementing culturally sensitive approaches to genetic counseling is essential. These strategies should ensure that families from diverse backgrounds can fully understand the implications of a positive newborn screen. (c) Providing families with comprehensive information and support is vital to help them make informed decisions regarding follow-up care and potential treatment options. (d) Ensuring that screening results are communicated clearly and promptly to both healthcare providers and families is critical for timely intervention. By addressing these areas, the newborn screening process for Gaucher disease can be optimized, leading to better understanding of screening for families of screen-positive newborns.

In conclusion, newborn screening for Gaucher disease has proven effective in identifying many newborns with the condition, allowing for early treatment in some cases. The success of the screening in New Jersey highlights the benefits of newborn screening for Gaucher disease. However, challenges remain, particularly with false positives and parental refusal to pursue confirmatory testing and follow-up care. These issues need to be addressed to enhance the effectiveness of newborn screening for Gaucher disease.

## Figures and Tables

**Table 1 IJNS-11-00034-t001:** Clinical Data for Confirmed and Suspected Cases of Gaucher Disease.

Subject	NBS Enzyme	Confirmatory Enzyme	Lyso-Gb1	Genotype	Diagnosis	Treatment	Age at Treatment Initiation
1	9.4% [≥12.0%]	0.2 nmol/h/mg Prot [≥8.7]	0.104 nmol/mL [≤0.040]	*GBA* c.1226A>G (p.N409S) homozygous	Type I	Y	42 months
2	<5.3% [≥12.0%]	0.49 nmol/h/mg Prot [≥3.53]	0.103 nmol/mL [≤0.040]	*GBA* c.635C>G (p.S212*)/*GBA* c.1226A>G (p.N409S)	Type I	Y	29 months
3	8.1% [≥12.0%]	0.77 umol/L/h [≥1.60]	0.550 nmol/mL [≤0.040]	*GBA* c.1226A>G (p.N409S)/*GBA* c.84dupG (p.L29fs)	Type I	Y	28 months
4	11.6% [≥12.0%]	1.08 nmol/h/mg Prot [≤3.53]	0.066 nmol/mL [≤0.040]	*GBA* c.1226A>G (p.N409S) homozygous	Type I	N	N/A
5	9.3% [≥12.0%]	1.04 nmol/h/mg Prot [≤3.53]	0.113 nmol/mL [≤0.040]	*GBA* c.1226A>G (p.N409S)/*GBA* c.1448T>C (p.L483P)	Type I	N	N/A
6	5.6% [≥12.0%]	0.44 umol/L/h [≥1.60]	32.47 ng/mL [<17.41]	*GBA* c.1226A>G (p.N409S) homozygous	Type I	N	N/A
7	<5.3% [≥12.0%]	1.14 nmol/h/mg prot [≥3.53]	0.059 nmol/mL [≤0.040]	*GBA* c.1226A>G (p.N409S) homozygous	Type I	N	N/A
8	10.3% [≥12.0%]	0.5 [4.0–22.6 nmol/h/mg]	Not Performed	*GBA* c.1226A>G (p.N409S) homozygous	Type I	N	N/A
9	3.8% [≥12.0%]	0.774 nmol/h/mg prot [7.5–14.5]	Not Performed	*GBA* c.1448T>C (p.L483P)/*GBA* c.680A>G (p.N227S)	Type I	N	N/A
10	5.3% [≥12.0%]	0.4 nmol/h/mg [4.6–12]	Not Performed	*GBA* c.1226A>G (p.N409S) homozygous	Type I	N	N/A
11	4.9% [≥12.0%]	Not Performed	Not Performed	*GBA* c.1226A>G (p.N409S) homozygous	Type I	N	N/A
12	< 5.3% [≥12.0%]	Not Performed	Not Performed	*GBA* c.1226A>G (p.N409S) homozygous	Type I	N	N/A
13	9.0% [≥12.0%]	0.618 nmol/h/mg [7.5–14.5]	12 ng/mL [<1]	*GBA* c.84dupG (p.L29fs)/*GBA* c.1226A>G (p.N409S)	Type I	N	N/A
14	8.9% [≥12.0%]	Not Performed	Not Performed	*GBA* c.1226A>G, p.(N409S) homozygous	Type I	U	N/A
15	8.8% [≥12.0%]	1.06 nmol/h/mg Prot [≤3.53]	0.031 nmol/mL [≤0.040]	*GBA* c.1226A>G (p.N409S)/*GBA* c.1148G>A (p.G383D)	Suspected Type I	N	N/A
16	8.9% [≥12.0%]	0.93 umol/L/h [>1.60]	9.26 ng/mL [<17.41]	*GBA* c.1226A>G (p.N409S)/*GBA* c.686C>T (p.A229V)	Suspected Type I	U	N/A
17	<5.3% [≥12.0%]	0.31 umol/L/h [≥1.60]	>200 ng/mL [<17.41 ng/mL]	*GBA* c.203del (p.P68fs)/*GBA* c.1448T>C (p.L483P)	Type II	Y	1 month
18	5.3% [≥12.0%]	0.48 nmol/h/mg Prot [≥3.53]	Not Performed	*GBA* c.1448T>C (p.L483P) homozygous	Type II	Y	13 months

NBS results, baseline confirmatory enzyme, Lyso-Gb1 levels, and genotype are presented in this table. Whether or not the individual is on treatment is based on the last clinical evaluation, with age at treatment initiation listed for those on enzyme replacement therapy (ERT). Treatment statuses for Subject 14 and Subject 16 are unknown, as these individuals no longer follow up with the reporting referral center. Key: Y: yes, N: no, and U: unknown.

**Table 2 IJNS-11-00034-t002:** Laboratory evaluations for Subject 1.

	DOL 14	24 Months	30 Months	36 Months	41 Months	46 Months	52 Months	62 Months
Lyso-Gb1 [≤0.040 nmol/mL]	0.104	0.194	0.198	0.372	0.504	0.199	0.171	0.109
Chitotriosidase [4–120 nmol/h/mL]	--	--	--	--	--	--	--	--
WBC [4.9–13.2 K/uL]	NC	7.5	6.8	8.2	7.0	8.1	6.9	5.4
RBC [3.90–5.30 10^6^/uL]	NC	4.29	4.12	4.21	4.23	4.07	4.38	4.33
Hgb [11.5–13.5 g/dL]	NC	10.6	10.9	11.2	11.3	11.1	12.1	12.3
PLT [150–450 10^3^/uL]	NC	263	156	162	189	234	235	266

Laboratory values are presented here for Subject 1. Subject 1 does not produce chitotriosidase. Biomarkers improved with treatment initiation at 42 months. WBC = white blood cells, RBC: red blood cells, Hgb: hemoglobin, PLT: platelets, and NC = not collected.

**Table 3 IJNS-11-00034-t003:** Laboratory evaluations for Subject 2.

	DOL 12	9 Months	16 Months	17 Months	19 Months	28 Months	34 Months	43 Months	49 Months
Lyso-Gb1 [≤0.040 nmol/mL]	0.103	0.104	0.749	NC	0.780	0.933	0.173	0.102	0.165
Chitotriosidase [4–120 nmol/h/mL]	NC	NC	8721	NC	10,486	9965	5239	1033	NC
WBC [5.1–13.4 K/uL]	NC	5.7	7.0	10.7	6.0	2.7	3.7	4.8	4.3
RBC [3.90–5.30 10^6^/uL]	NC	4.28	4.56	4.12	4.23	3.7	3.84	4.04	4.19
Hgb [11.5–13.5 g/dL]	NC	11.2	11.2	9.5	9.5	8.1	10.0	11.0	11.2
PLT [150–450 10^3^/uL]	NC	197	214	202	214	103	193	198	181

Laboratory values are presented for Subject 2. Subject 2 demonstrated increasing lyso-Gb1 and chitotriosidase prior to treatment initiation with improved values after initiation at 29 months of age. WBC = white blood cells, RBC: red blood cells, Hgb: hemoglobin, PLT: platelets, and NC = not collected.

**Table 4 IJNS-11-00034-t004:** Laboratory evaluations for Subject 3.

	26 Months	28 Months	30 Months	34 Months
Lyso-Gb1 [≤0.040 nmol/mL]	0.550	0.567	0.142	0.152
Chitotriosidase [4–120 nmol/h/mL]	8714	11,208	4264	1380
WBC [5.1–13.4 K/uL]	6.7	6.8	9.1	8.0
RBC [3.90–5.30 10^6^/uL]	4.71	4.75	4.99	5.30
Hgb [11.5–13.5 g/dL]	9.1	9.1	10.1	11.3
PLT [150–450 10^3^/uL]	126	155	183	227

Laboratory values are presented for Subject 3. There is a notable reduction in biomarkers after treatment initiation at 28 months. WBC = white blood cells, RBC: red blood cells, Hgb: hemoglobin, PLT: platelets, and NC = not collected.

## Data Availability

The data presented in this study are available from the corresponding authors upon request.

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
