# Peer review of "Newborn Screening for Gaucher Disease: The New Jersey Experience"

_2409-515X, 2025, doi:10.3390/ijns11020034_

Round 1
Reviewer 1 Report
Comments and Suggestions for Authors
Author reported their experiences about Newborn screening for Gaucher disease, in their New Jersy Experience. In a July 2019-Dec 2023, 438,515 newborns were screened with 60 screen-positive. Among them, 14 GD type 1, 2 GD type 2 were diagnosed. Three GD1 babies started ERT between 28-42 months while 2 GD2 babies started ERT since 1 and 13 months. Below are comments.
- Materials and methods: Page 3. 2.2 New Jersey Newborn Screening Protocol. Could author describe more about the assay in detail? Also, it will be great if can provide the assay result distributions as a figure in result part with highlighting the level differences among screen negative, true positive and false positive cases.
- Does the screening center use GBA to another enzyme activity ratio to decrease the false positive?
- Table 1. Case 9. The unit of enzyme activity “3.8” is missing. Genotype case 3 c.84dupG and case 17 c.203del, please add aminoacid change.
Author Response
- Materials and methods: Page 3. 2.2 New Jersey Newborn Screening Protocol. Could author describe more about the assay in detail? Also, it will be great if can provide the assay result distributions as a figure in result part with highlighting the level differences among screen negative, true positive and false positive cases.
Additional information regarding the assay is added. See lines 108-115 on page 3.
- Does the screening center use GBA to another enzyme activity ratio to decrease the false positive?
Additional information regarding the assay is added. See lines 108-115 on page 3.
- Table 1. Case 9. The unit of enzyme activity “3.8” is missing. Genotype case 3 c.84dupG and case 17 c.203del, please add amino acid change.
Corrected. See Table 1 Case 9, 3, and 17.
Reviewer 2 Report
Comments and Suggestions for Authors
This is a very important paper that describes the experience of NJ with NBS for Gaucher disease.
The number of samples tested is very impressive.
From smaller revisions to more important concerns:
-Always make sure the gene is written in italics. Page 3 line 102 & 117; page 10, lines 280-282
-Why was DBS lysogb1 not done as second-tier testing? This lack of second-tier testing leads to false positive rates, and high parental testing refusal. This lack of confirmation only increases parental anxiety.
- Out of the 438,515 newborns tested 2 were performed out of state for confirmatory and follow-up, 19 denied or were lost to follow-up, and 18 were Gaucher (but were the 18 out 39?) if so were the remaining 21 confirmed as false-positives? And how?
-Table 1, make it consistent if for enzyme activity you are using a cutoff that means affected < 12 or not affected > 12. It is not consistent for subjects 1 & 2.
-What was the matrix for lysogb1 testing? DBS or plasma? If DBS, the same DBS as the newborn screening or a new sample? It would have been crucial for lysogb1 to have been tested in the same dbs as second-tier. Also, even that different labs ran lysogb1 if the matrix was the same (e.g. DBS) how was one lab reporting a normal range of < 0.040 and the other <17.41. Even if the lab that ran 0.040 had this in plasma, this still seems very very low, even for plasma.
-Why did it take so long for the patients to start having treatment? If this is a NBS approach, the whole idea is to start treating the patients before symptoms arise.
-Was any ethnicity data recorded? This is important due to the higher prevalence of some variants in some genetic backgrounds.
-The clinical follow-up resulted in refusal of lysogb1. While if lysogb1 have done in the same dbs from the 1st tier-testing as second-tier testing the rates of parental refusal would have likely dropped.
-Write the meaning of each abbreviation on tables on the table footnote (e.g. WBC- white blood cells)
-Describe why and how Gaucher screening got added in NJ. Was is it by law? Is it done voluntarily like screenplus? Or mandated? And did this interfere with how the algorithm for testing was designed? Why was second tier not added?
-Page 9 line 261, the incidence provided is an approximation. The word "approximate" incidence should be added, because considering that several newborns were not tested due to refusal, loss of follow-up, death; this incidence is not the true incidence based on the 60 screen positives with appropriate testing after the screen positive.
-How were treatment decisions made?
-It will be important to keep track of gestational age, not only for Gaucher but also for Krabbe that is already screened in NJ.
-It is not correct to say that a two-tier approach should be enzyme testing followed by sequencing because of VUS. Anytime a biomarker is available, it is preferred to have the biomarker as 2nd tier testing, and molecular as 3rd tier testing. Really a 3 tier algorithm should be used.
-The sentence on page 10 lines 303-304 is not what the cited paper refers. Malsavia et al report that there is little data for pre-symptomatic patients, and they cite Gelb et al. Gelb et al do not say anything pre-symptomatic cases, they rather cite that biomarker is useful and cite other studies that have used this biomarker. Burlina et al., reports how useful dbs lysogb1 is as 2nd tier testing with newborns having elevated lysogb1 also having GBA variants. Only for one case they could not find the additional variant, but I am unsure if they tested for deletions, duplications, etc.
-Could the high parental refusal be for not having 2nd tier done before talking to the parents?
